# Gene Pathogenicity Prediction using Genomic Foundation Models

**Mohammad Amaan Sayeed[1,2], Hanan Aldarmaki[1], Boulbaba Ben Amor[2]**

[1]Mohamed Bin Zayed University of Artificial Intelligence
[2]Core42
{mohammad.sayeed;hanan.aldarmaki}@mbzuai.ac.ae, boulbaba.amor@core42.ai

## Abstract

The classification of pathogenicity in gene sequences plays an important role in deciphering genetic disorders and formulating precise medical treatments. Traditional methods for this classification task often involve an extensive analysis of several genomic attributes and complex predictive models, leading to a process that is both complex and computationally intensive. Recently, Large Language Models (LLMs), also known as Genomic Foundation Models, have been introduced, and their full potential in clinical applications is yet to be explored. In this work, we experiment with several such models, including HyenaDNA, GenaLM, and Nucleotide Transformer on the task of classifying pathogenic gene variants, benchmarking them against previous classification methods that rely on traditional feature extraction techniques. Our evaluation of fine-tuned models on the ClinVar dataset shows that the Nucleotide Transformer achieves an accuracy rate of 90%, which is on a par with some traditional pathogenicity prediction tools, yet it notably relies solely on genomic sequences, eschewing the need for additional data such as pathogenicity scores, conservation scores, or allele frequencies. These results indicate a potential for Genomic Foundation Models for a more streamlined and scalable gene sequence classification.

## Introduction

Genome sequencing is experiencing a rapid and dynamic evolution, revolutionizing our understanding of genetics and its impact on human health. Technological advancements have led to substantial growth in sequencing capacity while reducing costs, making it increasingly accessible (Qin 2019). This progress has allowed researchers to delve deeper into the intricate variations within our DNA. One of the most remarkable outcomes of this evolution is the identification of numerous genetic variants and their associated pathogenicity. Through comprehensive analyses, scientists can now unravel the genetic underpinnings of various diseases, enabling personalized medicine approaches and targeted therapies. Additionally, large-scale collaborative efforts, such as the Human Genome Project (Hood and Rowen 2013) and initiatives like the 100,000 Genomes Project (100,000 Genomes Project Pilot Investigators 2021), have contributed

to the compilation of extensive genetic databases, fostering a deeper understanding of rare and common genetic variations and their role in health and disease. However, due to present limitations, researchers are unable to examine the effects of each and every one of the approximately 20,000 rare missense variants in the human genome. Consequently, in order to help physicians and researchers assess the pathogenicity of missense variants, new technologies that are effective, scalable, and interpretable are required.

Supervised Learning approaches have shown promising results in the avenue of distinguishing pathogenic variants. Research on predictive algorithms has helped classify so many missense variants which otherwise would have been very difficult to classify (Evans et al. 2019). The recent work SNPred (Molotkov, Koboldt, and Artomov 2023) is an ensemble model specifically developed for predicting the pathogenicity of nonsynonymous single nucleotide variants (nsSNVs). This tool stands out for its ability to consistently outperform other state-of-the-art tools by combining so many pathogenicity prediction tools in an ensemble, especially when dealing with rare and cancer-related variants, as well as variants that are classified with low confidence by most tools. To characterize each variant, the authors utilized 33 pathogenicity prediction scores, 7 conservation scores, and 42 gnomAD and EXaC allele frequencies from dbNSFP as features. However, no existing technique uses the genome sequences as their direct input. Consequently, this creates a significant gap in our ability to comprehend the genome data in its pure, unprocessed state. The absence of direct genome sequence utilization hinders a more comprehensive understanding of the underlying genetic information and its true nature. To comprehend genomic sequences more effectively, foundational models tailored for genomic data are gaining prominence in research. One such model is DNABert (Ji et al. 2021), which utilizes a transformer-based architecture to generate embeddings for genome sequences. Nevertheless, its limitation in processing sequences longer than 500 tokens has posed challenges to its broader applicability in research. In response to this limitation, models like HyenaDNA (Nguyen et al. 2023), GenaLM (Fishman et al. 2023), and Nucleotide Transformer (Dalla-Torre et al. 2023) have emerged, each addressing the input length constraint in unique ways, thus expanding the scope and potential of genomic data analysis. These models have been pretrained on

vast and diverse genomic datasets, further enhancing their utility in understanding and analyzing genomic information. However, the utility of these genomic foundation models in impactful real-world applications has not been fully explored.

In this work, we fine-tune selected genomic foundation models, namely HyenaDNA, GenaLM, and Nucleotide Transformer on genomic sequences for pathogenicity prediction. We benchmarked these models using the filtered ClinVar (Landrum et al. 2014) dataset based on the state-of-the-art model SNPred (Molotkov, Koboldt, and Artomov 2023) criteria to ensure data reliability. Additionally, we showcase the embeddings produced by these models to demonstrate their problem understanding and capabilities in a visually interpretable manner. To the best of our knowledge, this study is the first to benchmark foundation models capabilities in gene pathogenicity prediction.

## Related Work

In this section, we start by examining the existing genomic and variant datasets. Following this, we discuss the traditional supervised learning techniques for gene pathoginicity prediction. The final section reviews some of the existing genomic foundation models used in this research.

### Genomic and Variant Datasets

Genome databases represent a cornerstone of modern biology, offering comprehensive repositories of genetic information. Among the most significant are the Human Genome (HG) builds (Nurk et al. 2022), which have mapped the entire human genome in successive iterations, each build enhancing the precision and completeness of the previous. These databases, like NCBI's GenBank (Benson et al. 2012) and the EMBL-EBI's Ensembl (Howe et al. 2021), provide not only the raw sequence data but also annotations and functional information. Integral to these resources are variant databases, which catalog genetic variations such as single nucleotide polymorphisms (SNPs) and structural variants. These databases, like dbSNP (Sherry et al. 2001) and ClinVar[1], are essential for understanding genetic diversity and disease associations. They enable researchers to correlate specific genetic variations with phenotypic traits, offering invaluable insights into human genetics, evolution, and personalized medicine. The synergy between HG builds and variant databases continually drives advancements in genomics, offering a comprehensive picture of the human genetic landscape. In particular, ClinVar has been used in developing and benchmarking models for predicting variant pathogenicity, a key factor in understanding genetic disorders and developing targeted treatments. One of ClinVar's most valuable features is its extensive collection of genetic variant data, each meticulously annotated with clinical significance. This includes detailed classifications of variants as benign, likely benign, uncertain significance, likely pathogenic, or pathogenic. These annotations are derived from a wide array of sources, including clinical laboratories, research institutions, and literature reviews, ensuring a rich and diverse dataset. The comprehensiveness of ClinVar's data makes it a suitable reference for the development and validation of computational algorithms designed to predict the pathogenicity of genetic variants. These algorithms, often based on machine learning or bioinformatics techniques, can help navigate the complexity of genomic data. They seek to automatically distinguish harmful mutations from benign genetic variations, a task crucial in clinical genetics and personalized medicine. By benchmarking these algorithms against ClinVar's well-curated and clinically annotated database, researchers can rigorously test the accuracy and reliability of their predictive models. This process ensures that these tools can accurately identify potentially disease-causing variants, a task that has direct implications for patient diagnosis and treatment planning. Furthermore, the continuous updating of ClinVar with new data and revised classifications allows for the ongoing refinement of these algorithms. As our understanding of genetic variants evolves, so too can the predictive models, ensuring they remain aligned with the latest clinical insights and genomic research.

### Pathogenicity Prediction Models

(Samocha et al. 2014) pioneered pathogenicity prediction in genomics, particularly for diseases like autism spectrum disorders (ASDs) with complex genetic factors. It introduced a statistical framework to analyze de novo mutations (DNMs) and identified key genes related to ASD, shedding light on neurodevelopmental processes. However, their work only tackled the problem at coding regions while 98.5% of the Human Genome is made up of non-coding regions. Notable approaches like CADD (Rentzsch et al. 2019) emerged as standards for identifying pathogenicity in non-coding sequences of the genome, although their accuracy faced challenges. GAVIN (van der Velde et al. 2017) improved accuracy by gene-specific adjustments and introduced clear Pathogenic and Benign labels. FATHMM (Shihab et al. 2013) with an extended feature set (FATHMM-XF) provides highly accurate genome-wide SNV predictions with confidence scores for simplified interpretation and cautious classification. To further enhance accuracy, (Ioannidis et al. 2016) introduced REVEL. It is an ensemble model that incorporates scores from multiple prediction tools, including namely MutPred (Mort et al. 2014), VEST (Carter et al. 2013), PolyPhen (Adzhubei, Jordan, and Sunyaev 2013) etc as its features. The current state of the art method SNPRed (Molotkov, Koboldt, and Artomov 2023) makes an ensemble combination of so many of the above listed methods. SNPred's evaluation involved using six distinct validation datasets derived from ClinVar and BRCA1 Saturation Genome Editing (SGE) data. The results showed that SNPred had a significant edge over other tools in various validation scenarios. Moreover, the study conducted by the authors of SNPred highlighted the limitations of using ClinVar data for evaluating the effectiveness of SNV pathogenicity tools, suggesting that such methods often lead to overstated performance estimates.

---

[1]https://www.ncbi.nlm.nih.gov/clinvar/intro/

| Model | Architecture | Pretraining | Size | Max. Context length | Resolution | Training Data |
|---|---|---|---|---|---|---|
| HyenaDNA Tiny | Stacked Hyena Operator layers | Next Nucleotide Prediction | 1M | 1,000 bp | Single Nucleotide token | Human Reference Genome |
| GenaLM Bert based large t2t | Transformer with BPE Tokenization | Masked Language Modeling | 110M | 4,500 bp | 3-mers to 6-mers | Human Reference Genome |
| Nucleotide Transformer 100M | Transformer | Masked Language Modeling | 100M | 12,000 bp | 6-mers | Human and other species |

Table 1: Comparison of best performing versions of the Foundation Models in terms of architecture, pretraining data, etc.

## Genomic Foundation Models

Recently, genomic foundation models such as HyenaDNA, GenaLM, and Nucleotide Transformer have been making headways in computational genomics. These models, drawing inspiration from breakthroughs in large-scale language processing, encode complex genomic sequences using self-supervised learning. Some details of the model variants used in this study are shown in Table 1.

**HyenaDNA.** HyenaDNA (Nguyen et al. 2023) is a genomic foundation model pretrained on the human reference genome. It extends context length up to 1 million tokens at single nucleotide-level, a significant leap over previous models. In addition, it scales sub-quadratically in sequence length, enabling faster training. Unlike previous models that used tokenizers to aggregate DNA sequences into larger units, HyenaDNA operates at single nucleotide resolution and incorporates full global context at each layer. It enables in-context learning for easy adaptation to new tasks without updating pretrained weights. HyenaDNA offers various model sizes to accommodate different computational needs and applications. These range from smaller versions like the `tiny-1k` and `small-32k` to larger ones like the `medium-450k` and the `large-1m`. The `large-1m` model, as the name suggests, can handle up to 1 million tokens. In benchmark applications, HyenaDNA has shown remarkable performance, achieving state-of-the-art results on 12 out of 18 basic genomic benchmarks, outperforming existing models in tasks like species classification and enhancer identification by significant accuracy margins.

**GenaLM.** GenaLM (Fishman et al. 2023) is a genomic foundation model inspired from natural language processing models like BERT and GPT. It is a transformer model with versions of both BERT and BigBird based architectures that undergo extensive pretraining on a vast array of unlabeled human genomic data, enabling it to encode complex genetic sequences and structures, much like the way a language model learns from text. It's versions of BERT and BigBird based models also have further last layer normalization and multi species pretraining data versions. In basic benchmark performances, GenaLM demonstrates its effectiveness in tasks like gene expression prediction, regulatory element identification, and genetic variant classification. Its ability to handle long genomic sequences with single nucleotide resolution gives it a substantial advantage in accurately identifying and interpreting genetic variations.

**Nucleotide Transformer.** Trained on an extensive dataset containing up to 174 billion nucleotides from various species, the Nucleotide Transformer (NT) (Dalla-Torre et al. 2023) is trained on Nvidia's Cambridge-1 supercomputer. This extensive training allows the model to handle a broad range of inputs and perform exceptionally across multiple benchmarks. The Nucleotide Transformer v2 notably advances over v1 by using a more extensive and diverse dataset, including 3,202 human genomes and 850 genomes from various species, enhancing its predictive capabilities for molecular phenotypes and includes models with 50m, 100m, 250m, 500m, and 2.5b parameters. In its performance evaluations, the Nucleotide Transformer was subjected to 19 different benchmarks to test its capabilities. Impressively, in 15 out of these 19 benchmarks, it either matched or exceeded the performance of other models that were specifically trained for those tasks. This level of performance not only demonstrates the model's versatility but also its effectiveness in translating DNA sequences into RNA and proteins. One of the key strengths of the Nucleotide Transformer is its ability to focus attention on crucial genomic elements, such as enhancers that regulate gene expression, without any direct supervision. This capability is particularly valuable for the accurate prediction of molecular phenotypes from DNA sequences alone.

## Experiments and Results

In this work, the ClinVar dataset served as the source for variant information based on the Hg38 human genome assembly. Following the SNPred paper's experimental settings, we compiled a dataset containing 50,000 instances of both Pathogenic and Benign Variants. For example, for the HyenaDNA Tiny model, we extracted subsequences spanning 1,000 nucleotides – 500 on either side of each of the variant – in alignment with the input size of the model. The construction of our test set also follows the protocols described in (Molotkov, Koboldt, and Artomov 2023), incorporating variants added to ClinVar post-April 2022, while our training set includes data prior to this date. Our early experiments show that performance is highly sensitive to the balance of samples per class in the training set. As all the samples were already selected based on their ClinVar rating, we randomly downsample the class with excess samples to balance the training set. Regarding the experimental setup, we fine-tuned 10 models from each of the three foundational model families: HyenaDNA, GenaLM, and NT. Given the

| Model | Accuracy | Precision | Recall |
|---|---|---|---|
| HyenaDNA Tiny | 56.65 | 66.59 | 43.37 |
| GenaLM Bert t2t | 60.58 | 81.56 | 69.52 |
| **GenaLM Bert based lastln t2t** | 67.81 | **94.80** | 79.07 |
| GenaLM Bert Large t2t | 68.74 | 87.16 | 76.86 |
| GenaLM BigBird base t2t | 66.03 | 89.51 | 76.00 |
| Nucleotide Transformer 50m so | 88.54 | 89.83 | 89.18 |
| **Nucleotide Transformer 100m so** | **90.62** | 91.17 | **90.89** |
| Nucleotide Transformer 250m so | 89.46 | 91.25 | 90.35 |
| Nucleotide Transformer 500m so | 78.24 | 82.15 | 80.15 |

Table 2: Fine-tuned models performances (Accuracy (%), Precision (%) and Recall (%).)

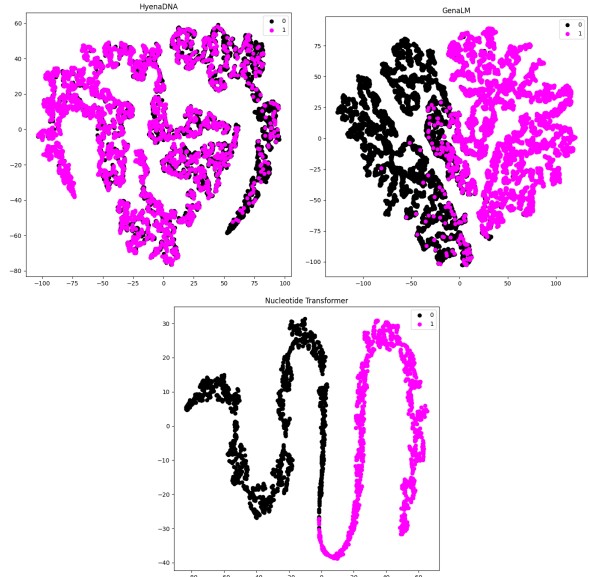

Figure 1: t-SNE plots of embeddings from the best fine-tuned variant of each model (O:Benign; 1:Pathogenic).

compute constraints, we could only fine tune the tiny version of HyenaDNA model family while we covered most of the models from the other two families. The fine-tuning process utilized default hyperparameters as outlined in their respective publications. To assess the models, we employed the average accuracy, precision and recall an evaluation metrics. We summarize these results in Table 2.

NT achieved the best overall performance in terms of accuracy and recall, while GenaLM models achieved higher precision. We also see that the NT outperforms the other model families in terms of AUC ROC and AUC PR scores, as shown in Table 3. The table also shows, in order of decreasing performance, various other models previously reported on this test set, reproduced from (Molotkov, Koboldt, and Artomov 2023). While none of the LLMs reached the performance of the best baselines, NT outperformed roughly half of the traditional models, whereas the other models we tested lagged behind. One possible reason for NT's relatively good performance in this benchmark is the fact that it was massively pre-trained on over 3200 diverse human genomes which helped it outperform so many other models on other genome benchmark datasets. Figure 1 shows

| Model | AUC ROC | AUC PR |
|---|---|---|
| SNPred | 0.994218 | 0.993159 |
| bayesdel.add_af | 0.986162 | 0.985381 |
| metarnn | 0.977834 | 0.936478 |
| clinpred | 0.973365 | 0.924906 |
| cadd | 0.963422 | 0.959961 |
| revel | 0.961136 | 0.902031 |
| mvp | 0.943756 | 0.856274 |
| eigen | 0.931853 | 0.92528 |
| deogen2 | 0.93068 | 0.853016 |
| m-cap | 0.925593 | 0.799451 |
| metasvm | 0.921943 | 0.80297 |
| vest4 | 0.919821 | 0.891856 |
| mutpred | 0.919708 | 0.876561 |
| metalr | 0.917977 | 0.817804 |
| mutationassessor | 0.909111 | 0.795324 |
| polyphen2.hvar | 0.896462 | 0.737596 |
| **Nucleotide Transformer 100M** | **0.892019** | **0.851482** |
| sift | 0.880152 | 0.721437 |
| sift4g | 0.873966 | 0.692378 |
| polyphen2.hdiv | 0.873389 | 0.717018 |
| mutationtaster | 0.869145 | 0.878079 |
| list-s2 | 0.86067 | 0.650734 |
| primateai | 0.848458 | 0.611447 |
| mpc | 0.818088 | 0.626343 |
| fathmm-mkl | 0.80056 | 0.736335 |
| dann | 0.775147 | 0.63381 |
| lrt | 0.745428 | 0.719936 |
| **GenaLM Large** | **0.7318050** | **0.717726** |
| fathmm-xf | 0.684203 | 0.606642 |
| genocanyon | 0.655255 | 0.640234 |
| **HyenaDNA tiny** | **0.608079** | **0.477324** |
| h1-hesc | 0.604735 | 0.637964 |
| gm12878 | 0.602584 | 0.636317 |
| huvec | 0.601604 | 0.63633 |
| integrated | 0.601414 | 0.632413 |

Table 3: Comparative performances of various models based on AUC ROC and AUC PR.

t-SNE plots representing the embeddings obtained from the best fine-tuned variant of each model family. The plots show a clear separation between phathogenic and benign variants using the NT model, and to some extent the GenaLM model, which is consistent with their classification performance.

## Conclusion

We presented in this paper the first benchmarking study of three different Genomic Foundation Models in predicting gene pathogenicity. Namely, we tested the HyenaDNA, GenaLM, and Neucleutide Transforemer (NT). The ability of each model to interpret genomic data is significantly influenced by its preliminary training, its context length, and resolution. Using these genomic foundation models enables pathogenicity analysis directly from genome sequences. Compared to previous pathogencity prediction models, our experiments show that the NT is ranked in the 50th percentile, outperforming roughly half of the traditional methods. While still lagging segnificantly behind the state-of-the-art methods, these results show some potential for genomic foundation models to tackle challenges associated with human genomic data analysis.

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
