# OpenReview forum: "Gene Pathogenicity Prediction using Genomic Foundation Models"
_AAAI.org/2024/Spring_Symposium_Series/Clinical_FMs — AAAI 2024 SSS on Clinical FMs_

### Official Review · Reviewer_n948 · 2024-02-19
**Comparison of Genomic Foundation Models**

**Rating:** 5
**Confidence:** 2

**Review:**

## Summary

This paper compares three Genomic Foundation Models (HyenaDNA Tiny, GenaLM, and Nucleotide Transformer) against each other on the ClinVar dataset for gene sequence classification. Then the best model is compared against the state-of-the-art traditional genomic model (SNPred). I do not work in the genomics space; I work in adjacent domains (healthcare, language models), so I can see that the comparisons the authors are trying to make may be valuable to other genomics researchers. However, this manuscript reads more like a review paper than one describing new research. It is missing key details of experiments and a clear description of fine-tuning and evaluation datasets is missing, making it hard to interpret results.

## Pros
* Well written detailed description of prior work in this domain
* Detailed metrics comparison of transformer-based foundational models in genomics against state-of-the-art traditional models like SNPred.
* tSNE embedding visualization helps visually explain performance gap between models

## Cons
* Too much of the manuscript is focused on introduction/background/related work and too little on experiment and discussion. Some of the background is helpful to readers not in the genomic space, but overall should be reduced by at least 1 page to make space for experiment and broader discussion of results. Otherwise, this submission reads like a review paper rather than an original experiment paper.
* Nit pick: Formatting of text in Table 1 is suboptimal. To make it more readable, consider using abbreviations and describe abbreviations in table legends.
* Nit pick: Formatting of Table 3 is odd. There are 6 significant figures reported for SNPred and the other models, but only 2 significant figures for Nucleotide Transformer. Be consistent.
* Table 2 reports comparison of three Genomic Foundational models using Accuracy, Precision, Recall, and then the best one (Nucleotide Transformer 250M) is compared against SNPred in Table 3 using AUC ROC and AUC PR.  It seems like it would be more straightforward and clearer to just report AUC ROC and AUC PR for all 3 models in a single table along with SNPred.
* Authors describe that 10 models were fine-tuned, but then only report a single result per model. It is unclear whether there was further model selection from the 10 models or if evaluation results were aggregated, etc.
* Larger HyenaDNA models were not compared due to compute resource constraints.
* Surprising to me that Nucleotide Transformer 500M performs worse than Nucleotide Transformer 250M and there is no discussion of why this is the case.
* Fine-tuning and Evaluation datasets are not well described. Would recommend a table that describes pertinent statistics for each dataset and split that is used.

---

### Official Review · Reviewer_9kk6 · 2024-02-19

**Rating:** 7
**Confidence:** 3

**Review:**

The paper presents the first benchmarking study of three different Genomic Foundation Models in predicting gene pathogenicity. The interpretive capacity of each model is significantly affected by factors such as preliminary training, context length, and resolution. Conventional approaches have limitations in enhancing model interpretation and heavily rely on specific features. The study underscores the potential of Foundation Models to address the complex challenges associated with genomic data analysis.

Quality and clarity:
- The quality is clear, and the tables and figures are generally understandable. Small notes: Figure one axis should either be made larger or removed, Tables could have standard errors
- The paper is generally well-written and free of grammatical errors

Originality and significance:
- Nucleotide Transformer exhibits better generalizability and accuracy performance compared to other models.
- The excellent performance of Nucleotide Transformer can be attributed to its extensive pretraining on over 3200 diverse human genomes, which sets it apart from other models on various genome benchmark datasets.
- Nucleotide Transformer solely utilizes genomic sequences for gene pathogenicity prediction, whereas other approaches rely on different features.

Weaknesses
- Limited Explanation of Embeddings, as they fail to properly distinguish between pathogenic and benign variants. How can this be addressed?
- Lack of Comparison with Non-LLM Models: The study primarily focuses on comparing different Genomic Foundation Models and LLMs. However, a comprehensive evaluation should also include comparisons with non-LLM-based approaches to assess the relative advantages and disadvantages of using LLMs for gene pathogenicity prediction.
- Computational limitations could be addressed by LORA or other efficient NLP techniques

---

### Official Review · Reviewer_NXqN · 2024-02-21
**Gene Pathogenicity Prediction using Genomic Foundation Models**

**Rating:** 7
**Confidence:** 3

**Review:**

This paper analyzes Genomic Foundation Models' effectiveness in gene pathogenicity prediction, highlighting Nucleotide Transformer's superior performance. It showcases the potential of direct genomic sequence classification, offering a significant advancement over traditional methods. The study is innovative and has a practical impact but lacks broad model evaluation, external validation, and discusses potential biases minimally.

**Pros**
- Innovative Approach: The paper pioneers in evaluating Genomic Foundation Models for gene pathogenicity prediction, showcasing a significant shift from traditional methods.
- Comprehensive Benchmarking: It rigorously benchmarks across different models, providing a detailed comparison of their performance on a widely recognized dataset.

**Cons**
- Model Specificity: The study's focus on a limited number of models might not capture the full spectrum of potential genomic foundational models.
- Dataset Limitations: Reliance on the ClinVar dataset alone could introduce bias or limit the generalizability of the findings across different genomic contexts.
- Lack of External Validation: The study does not provide external validation of the models' performance on datasets outside of ClinVar, which could further establish their utility and robustness.

---

### Official Review · Reviewer_DfKm · 2024-02-22
**This paper presented the first benchmark study of genomic foundation models, which provides insights for future investigation.**

**Rating:** 5
**Confidence:** 3

**Review:**

Advantages:
1. Gene pathogenicity prediction is a significant task, which can help us understand genetics and its impact on human health.
2. A benchmark study is an urgent need of this research community.

Disadvantages:
1. This paper focuses on three models: HyenaDNA, GenaLM, and Nucleotide Transformer. However, these models are relatively small.
2. This paper is far away from "comprehensive", which is important for benchmark study. There are few datasets, tasks and models for evaluation.